# Super-resolution kinetochore tracking reveals the mechanisms of human sister kinetochore directional switching

Nigel J Burroughs[1]*[†], Edward F Harry[2][†], Andrew D McAinsh[3]*

[1]Warwick Systems Biology Centre, Warwick Mathematics Institute, University of Warwick, Coventry, United Kingdom; [2]Warwick Molecular Organisation and Assembly in Cells, University of Warwick, Coventry, United Kingdom; [3]Centre for Mechanochemical Cell Biology, Division of Biomedical Cell Biology, Warwick Medical School, University of Warwick, Coventry, United Kingdom

**Abstract** The congression of chromosomes to the spindle equator involves the directed motility of bi-orientated sister kinetochores. Sister kinetochores bind bundles of dynamic microtubules and are physically connected through centromeric chromatin. A crucial question is to understand how sister kinetochores are coordinated to generate motility and directional switches. Here, we combine super-resolution tracking of kinetochores with automated switching-point detection to analyse sister switching dynamics over thousands of events. We discover that switching is initiated by both the leading (microtubules depolymerising) or trailing (microtubules polymerising) kinetochore. Surprisingly, trail-driven switching generates an overstretch of the chromatin that relaxes over the following half-period. This rules out the involvement of a tension sensor, the central premise of the long-standing tension-model. Instead, our data support a model in which clocks set the intrinsic-switching time of the two kinetochore-attached microtubule fibres, with the centromeric spring tension operating as a feedback to slow or accelerate the clocks.

*For correspondence: N.J. Burroughs@warwick.ac.uk (NJB); A.D.McAinsh@warwick.ac.uk (ADM)

[†]These authors contributed equally to this work

**Competing interests:** The authors declare that no competing interests exist.

## Introduction

The accurate segregation of chromosomes during anaphase requires that all sister kinetochores bi-orientate, an attachment state in which sisters form stable attachments to the plus-ends of microtubules that originate at opposite spindle poles. Bi-orientation begins immediately after nuclear envelope breakdown during prometaphase when scattered chromosomes engage the nascent mitotic spindle, and concludes with the formation of the metaphase plate – a state where all sister kinetochores are bi-orientated and aligned on the equator of a bipolar spindle (*McIntosh et al., 2012*). To achieve this bi-orientation, sister kinetochores must be able to undergo directed movements to the equator (this is termed congression) and then maintain their position prior to anaphase onset – a feature of this latter phase is oscillations of the chromosomes along the spindle axis. Directed motility is possible because one sister adopts a poleward (P) moving state (the lead sister) while the other is in an away-from-the-pole (AP) moving state (the trailing sister). These two movement states reflect the balance of microtubule polymerisation/depolymerisation within the kinetochore-fibre (K-fibre), which is typically 20–25 microtubules in human cells (*Compton, 2000*; *Rieder, 2005*; *Wendell et al., 1993*). While such K-fibres are rarely coherent, there is a small polymerisation bias between sister kinetochores (*Armond et al., 2015*; *VandenBeldt et al., 2006*) and they can be thought of as being in either a net polymerising or depolymerising state. The adaptive switching between these AP and P states then defines the directionality of chromosome motion and can give rise to the quasi-periodic oscillations that are observed in the majority of vertebrate cells (*Skibbens et al., 1993*). An

**eLife digest** In human cells, DNA is arranged into structures called chromosomes. Before a cell divides it copies its entire set of chromosomes to make paired chomosomes known as sister chromatids. Then, the sister chromatids are separated to ensure that each new daughter cell contains a full set of chromosomes. A structure called the spindle is responsible for separating the sister chromatids. It is made of long filaments called microtubules that grow out from 'poles' at opposite sides of a cell. The two sister chromatids in each pair attach to microtubules that originate from opposite ends of the cell. This attachment is achieved by a protein machine called the kinetochore, which can move along microtubules as they grow or shrink.

Prior to the separation of the chromatids, the paired sister chromatids are moved into positions so that they are approximately an equal distance from the two poles. For the majority of the time, one sister kinetochore moves towards the pole it is attached to (called the lead sister), while the other sister moves away from the pole it is attached to (the trailing sister). Then, the kinetochores swap roles and move in the opposite direction. In most cells, the pairs of sister kinetochores repeatedly switch between moving backwards and forwards giving rise to oscillations in the positions of the sister chromatids.

It is not clear how the two sister kinetochores are able to communicate with each other so that they can co-ordinate their backwards and forwards movement. It is currently thought that the lead kinetochore changes direction first because of an increase in the distance between the two sisters, thereby increasing the tension between the sisters (like a spring being stretched).

Burroughs, Harry and McAinsh revisit this idea by looking at living human cells and tracking the movement of the kinetochores in great detail. Using computational techniques to analyse kinetochore movements in living cells, the experiments reveal that the trailing sister kinetochore can sometimes change direction before the lead sister. When this happens the sisters start to move apart until the previously leading sister switches direction, so that the sisters then movingve together in the same direction. The distance between the sisters remains high until the next time the sisters change direction, which means that 'tension' cannot solely be responsible for the communication between sister kinetochores.

Burroughs, Harry and McAinsh's findings suggest instead that sister kinetochores contain a 'clock' that decides when they will change direction. The tension between the two sister chromatids is still important, and acts to change the time of the clocks. The next challenge is to understand how these clocks work and which parts of the kinetochore are involved.

outstanding question is to understand the mechanisms by which the two sister kinetochores are able to communicate in order to coordinate their P/AP states and thereby generate chromosome movements.

Initial investigations into the control of chromosome movement utilised time-lapse imaging in newt lung cells using video-enhanced differential interference contrast microscopy (*Skibbens et al., 1993*). Kinetochores were shown to undergo periods of relatively constant velocity separated by abrupt changes in direction – a behaviour termed 'directional instability'. Subsequent experiments demonstrated that weakening the centromeric chromatin which links the sisters (with a laser) uncoupled the normally coordinated motility of sister kinetochores (*Skibbens et al., 1995*). These experiments led to a model (also see *Rieder and Salmon, 1994*) in which tension in the centromeric chromatin triggers a lead sister switch (P-to-AP) at a certain threshold, the loss of tension then triggering a directional switch in the second sister (AP-to-P). More recent kinetochore-tracking experiments in PtK1 cells are consistent with this model and show that switching initiates at maximum inter-kinetochore stretch (*Wan et al., 2012*), schematic shown in *Figure 1*. The polar ejection force, which increases with proximity to the pole, pushes the chromosomes towards the metaphase plate. When chromosomes stray far from the equator, this anti-poleward force increases the load on the leading (P) kinetochore and promotes switching – an idea supported by experiments in newt and human cells (*Ke et al., 2009*; *Stumpff et al., 2012*). The standard tension model thus predicts a fixed sequence of sister kinetochore-switching events during a directional reversal – lead switch first,

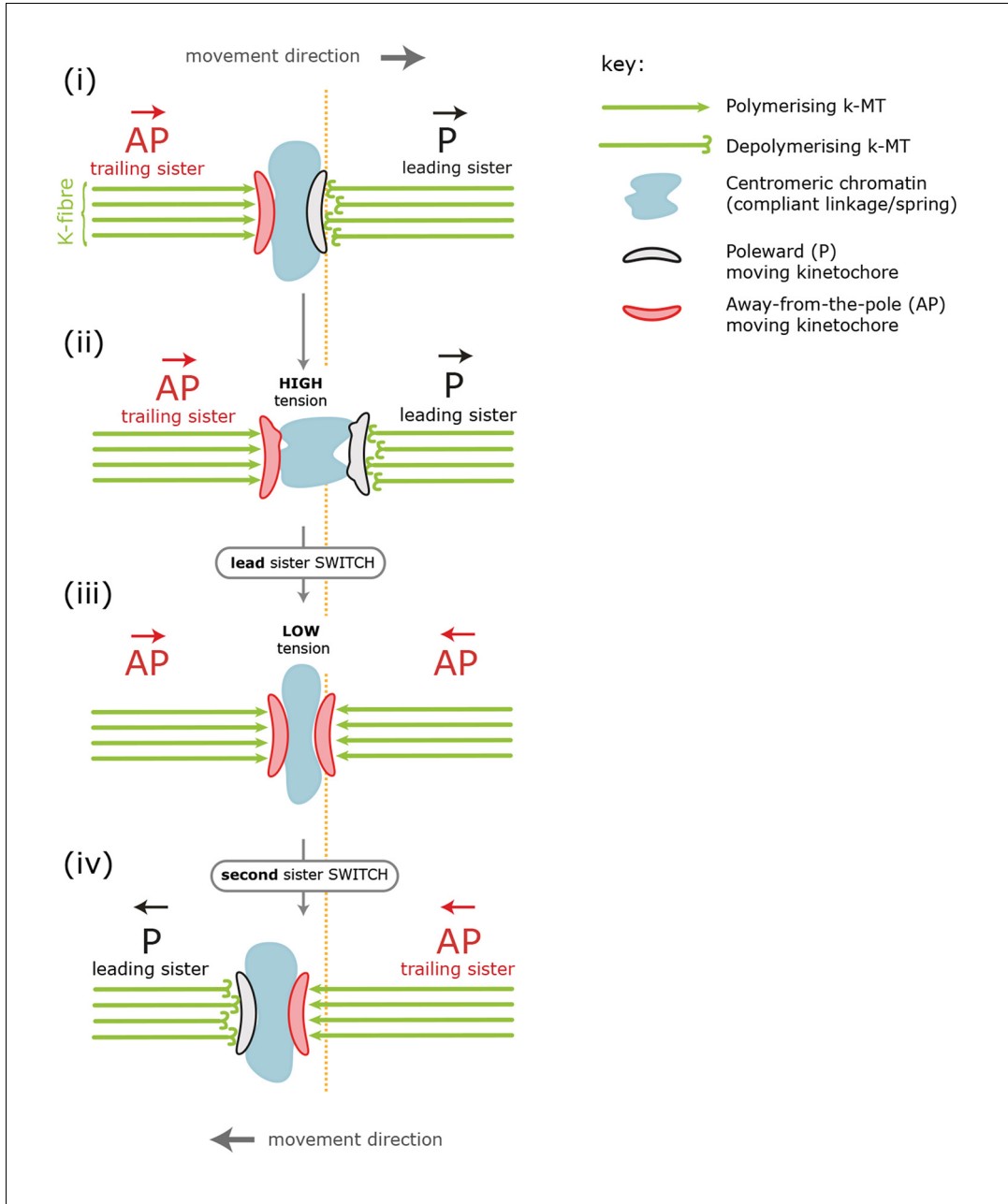

**Figure 1.** Standard model for kinetochore directional switching. Schematic outlining the prevailing model for how sister kinetochores coordinate directional switches. As the leading, poleward-moving, kinetochore (P; black) moves to the right, the centromeric chromatin (blue) – which functions as a compliant linkage between sisters – becomes progressively more stretched (steps i, ii). Stretching occurs because the trailing, away-from-the-pole, kinetochore (AP; red) is moving more slowly than the lead. Once sisters are at maximum stretch, the tension in the chromatin is thought to trigger the lead sister kinetochore to switch into an AP state. This results in a rapid loss of tension as both sisters are then in an AP state and moving towards each other (step iii). This relaxation is thought to trigger switching of the initially trailing sister into a P-moving state (step iv). Adapted from *Wan et al., 2012*.

followed by trail. However, the timeframe for these events is short (several seconds), requiring a sampling rate that avoids temporal aliasing. Existing kinetochore-tracking assays have a frame rate in the 7.5–15 s range (*Dumont et al., 2012*; *Jaqaman et al., 2010*; *Wan et al., 2012),* meaning that detailed analysis of the switching mechanism has not been possible to date.

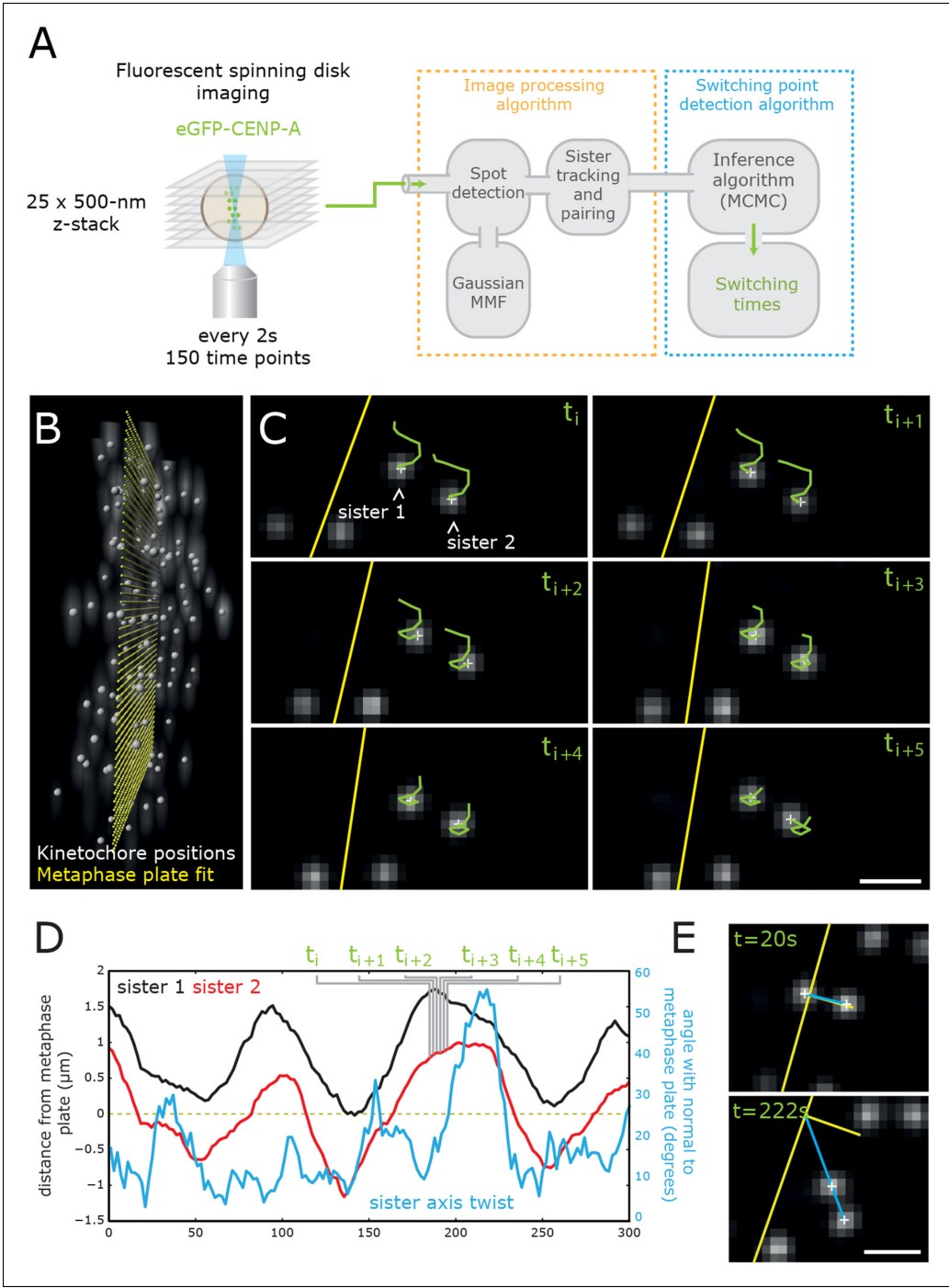

**Figure 2.** High-throughput tracking of kinetochores with sub-pixel spatial resolution. (**A**) Imaging and analysis flow chart summarising the steps of image sequence acquisition, image processing and statistical trajectory analysis. MMF- mixture model fitting, MCMC- Markov chain Monte Carlo algorithm. (**B**) 3D view of kinetochores (eGFP-CENP-A – spot location marked by white spheres) relative to the metaphase plate (yellow spheres). (**C**) 2D images (single Z plane) of a sister kinetochore pair over six frames. Green tails represent kinetochore trajectories over the previous six frames (12 s). Yellow line indicates the metaphase plate intersection with the image plane. Sub-pixel kinetochore positions marked with white crosses. Scale bar = 1 µm. (**D**) Full sister pair trajectory showing the (normal) distance from the metaphase plate of the two sisters (black and red), and the sister axis twist angle relative to the metaphase plate normal (blue). Frames indicated in green correspond to the image slices shown in (**C**). (**E**) 2D images (single Z plane) of the sister kinetochore pair in (C/D) at the minimum and maximum twist. Sister axis shown in blue, metaphase plate and its normal in yellow. Scale bar = 1 µm.

*Figure 2 continued on next page*

*Figure 2 continued*

The following figure supplement is available for figure 2:

**Figure supplement 1.** Sister distance autocorrelation and distribution.

## Results and discussion

We developed a high-resolution kinetochore-tracking procedure and used a switching point detection algorithm to examine the fine detail of paired sister kinetochore trajectory data. Our previous kinetochore-tracking assay (*Jaqaman et al., 2010*) with a 7.5-s sampling time lacks the time resolution to resolve the relative sister switching order. To improve resolution, we used spinning disk confocal microscopy to capture 3D image stacks every 2 s over 150 time steps in HeLa-K cells expressing a marker for the kinetochores (either eGFP-CENP-A or eGFP-CENP-A eGFP-Centrin1; *Figure 2B*; *Video 1*, *Video 2*). Phototoxicity was minimal, as >90% of cells successfully underwent anaphase. Sister kinetochores were tracked as previously described (*Jaqaman et al., 2010*), except that we implemented 3D Gaussian mixture model fitting for determining sub-pixel spot locations (*Thomann et al., 2002* see *Figure 2A, C*; *Videos 3–5*) – important here because our faster time sampling results in a smaller inter-frame spot displacement requiring higher localisation accuracy. This sub-pixel (super-resolution) tracking gives high theoretical positional accuracy ($x,y = \pm 2.8$ nm; $z = \pm 5.7$ nm; see 'Materials and methods') and reveals kinetochore dynamics in exquisite detail (*Figure 2D*). Consistent with previous work (*Jaqaman et al., 2010*; *Vladimirou et al., 2013*), sister kinetochores had a mean inter-kinetochore distance of ~910 nm and underwent quasi-periodic oscillations normal to the metaphase plate with a half-period of 35 s (*Figure 2—figure supplement 1A, B*). Finally, we constructed a Bayesian switching point inference algorithm that estimates from an observed sister pair trajectory the switching times for each sister (most probable frame) and the directional switching events by assignment of a direction of movement to each sister (see 'Materials and methods'). Here, we focus on *coherent runs* (periods when the sisters are moving in the same direction) and the switching events that end runs. We tested this algorithm on simulated data where the true switch time is known giving accuracies of 94% (see 'Materials and methods'; *Figure 3A* and *Figure 3—figure supplement 1*). This switching point algorithm determined whether the leading or trailing sister switches first in a directional reversal of the sister pair and by how many frames.

Next, we identified switch events in the trajectories of 1529 sister pairs across 55 eGFP-CENP-A cells and calculated the frequency that lead or trailing sisters switch first (n=9022 events ending a coherent run). We note that these trajectories were distributed throughout the metaphase plate (*Figure 3B*). We defined a directional switching event as a lead initiated directional switch (LIDS) if the lead sister switched at least one frame before the trailing sister and a trail initiated directional switch (TIDS) similarly. The remaining events correspond to sisters switching within the same frame and are denoted joint directional switches (JDS, e.g. see *Figure 3C,D*). These criteria illustrate that there is a strong lead bias with the fraction of LIDS and TIDS being 54.3% and 34.8%, respectively, the remaining fraction (10.9%) of events being joint. These data demonstrate that trailing sister switching is suppressed in human cells, leaving a lead-to-trail bias of 1.56:1.

Current directional switching models propose that switching of the lead kinetochore is initiated when the inter-sister distance (centromere spring) reaches a maximum stretch (tension) (*Rieder and Salmon, 1994*; *Skibbens et al., 1995*; *Skibbens et al., 1993*; *Wan et al., 2012*). Moreover, no mechanism has been proposed to explain trail first switching; trailing sister initiated switching has also been reported for PtK1 cells at a 15% frequency (*Dumont et al., 2012*). However, as we show here, the existence of trail first switching has ramifications for both the sister-sister coupling and the possible switching control mechanisms. By aligning profiles of the inter-sister distance – which reflects tension in the centromeric chromatin, 40 s before and 40 s after the first sister switching event (*Figure 4A*), we demonstrate clearly that LIDS and TIDS both have strong pre-event and post-event inter-sister distance signatures (*Figure 4A*; compare red [TIDS] and black [LIDS] traces). These stereotypical dynamic-tension signatures at LIDS/TIDS events (discussed below) demonstrate that our assignment of a LIDS or TIDS (*Figure 3*) are robust and meaningful.

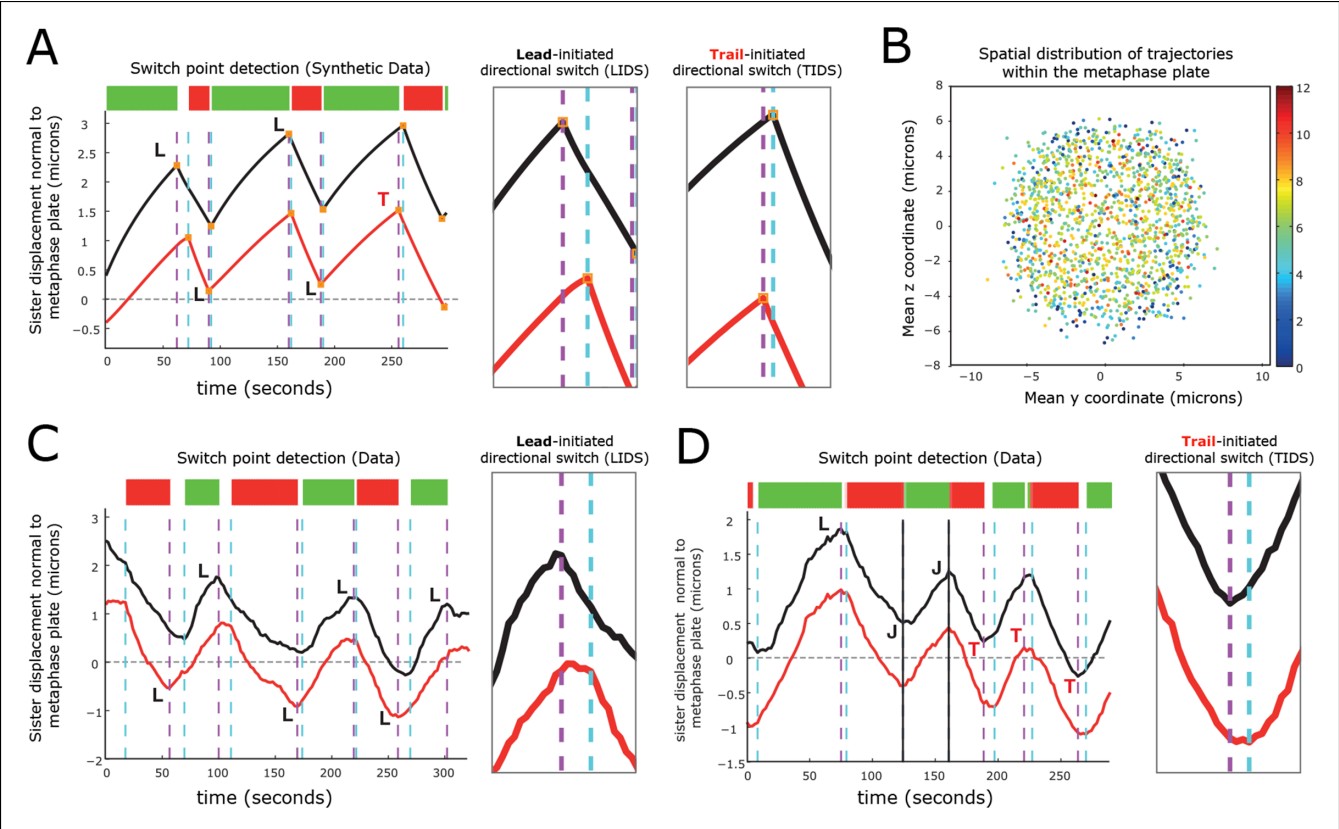

**Figure 3.** Kinetochore switching dynamics. (A) Event detection on a synthetic data set with idealised oscillation of two sisters (for parameters, see 'Materials and methods'). True switchings are shown as orange squares with true coherent state shown at the top of the figure. Green shows depolymerising/polymerising sister state (-/+), and red shows polymerising/depolymerising sister state (+/-), sister 1/sister 2, respectively. Detected switching events shown as vertical dashed lines: coherent run initiation/end shown in cyan/purple, respectively. The sister that initiates a directional switch is indicated: lead (L) initiated directional switch (LIDS), trail (T) initiated directional switch (TIDS) and joint (J) directional switch (JDS). Horizontal grey line indicates position of spindle equator. Enlargements of LIDS and TIDS events. (B) Trajectory positions within the metaphase plate viewed along the spindle axis (y,z). Colour indicates the number of events detected in the trajectory that end a coherent run, n=1529 trajectories from 55 cells. (C, D) Example trajectories from live cells showing detection of switching, colours as in (A) except inferred coherent state is shown at the top and joint switching events shown as vertical solid black lines.

The following figure supplement is available for figure 3:

**Figure supplement 1.** Autoregressive model simulation produces qualitatively realistic oscillations.

For a LIDS, the inter-sister distance increases corresponding to an average increase of spring stretch from 14% to 20% over the 20 s prior to the first sister switch, stabilises 4 s prior to that switch event and then decreases rapidly during the 6 s following the first sister switch to a minimum average stretch of 5% (*Figure 4A*; black trace). Here, we used a baseline rest length of 788 nm determined from nocodazole-treated cells (see 'Materials and methods'). The LIDS event leaves the two K-fibres in a polymerising state, which decreases the inter-sister distance; this is in line with the standard model where the inter-sister distance will then undergo extension over the following coherent run (*Figure 1*). The relaxation of the spring immediately prior to the first sister switch appears to be the result of the lead sister slowing down (data not shown). This would be in line with experiments in Ptk1 cells in which the velocity slows at maximal inter-sister stretch (*Wan et al., 2012*). Finally, the switching of the second (trailing) sister correlates with the reduction in spring extension to near zero (*Figure 4A*), possibly suggesting that switching is the result of a loss of tension in accord with the standard tension model (*Rieder et al., 1994*).

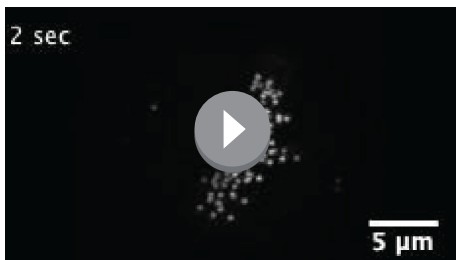

**Video 1.** Length: 5 s; Real Time: 300 s; Frame Rate: 30 fps. HeLa-K eGFP–CENP-A, eGFP– Centrin1. Z-projection through 12 μm. (Deconvolved) Movie of a metaphase cell (captured at 2 s per frame). Movie rendered used MATLAB and ImageJ. Please also refer to *Figure 2*.

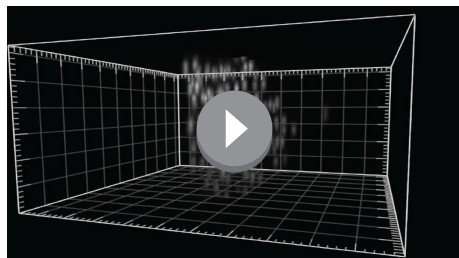

**Video 2.** Length: 16 s; Real Time: 300 s; Frame Rate: 30 fps. eGFP–CENP-A, eGFP–Centrin1. (Deconvolved) Movie of a metaphase cell rendered in 3D. Movie rendered used MATLAB and IMARIS. Please also refer to *Figure 2*.

However, the profile for a TIDS is distinct: the inter-sister distance is much lower prior to the first sister switch (*Figure 4A*; red trace). The maximum stretch (25%) is only reached 4 s after the switch has occurred. This indicates that the TIDS itself is necessary to build up tension in the centromeric chromatin while the maximum inter-sister stretch is higher than that seen during LIDS (average 200 nm spring extension). The LIDS and TIDS signatures are also present if we condition on the previous event type (*Figure 4B*), showing that although the previous event type does affect the inter-sister distance prior to the next event, the qualitative form is similar close to the switching event. The high overstretch of the centromeric chromatin during a TIDS raises a fundamental challenge to the standard tension model of kinetochore oscillations which states that the spring tension rises during a run of the sister pair (to the left or right) triggering a leading sister switch and relaxes during a directional switching event which triggers switching of the second sister (*Figure 1*). As illustrated above, this describes a repeated LIDS choreography. However, following a TIDS event, the centromeric spring tension escalates during the directional switch and starts to relax 4 s after the directional switch (*Figure 4A*), suggesting that the tension remains high over the following run.

To dissect this further, we examined the inter-sister distance over averaged coherent runs, (n=6339) by aligning the start and the end of the intervening run (rescaling time of each run to a standard length of 1) (*Figure 4C,D*). For runs with a preceding switch that was either a LIDS or a TIDS, the inter-sister distance rose (*Figure 4C*) or relaxed (*Figure 4D*) over the following run, respectively. At the end of the run, the spring profile had the LIDS or TIDS signature depending on the next event type (*Figure 4C,D*). Crucially, for runs starting after a TIDS the subsequent directional

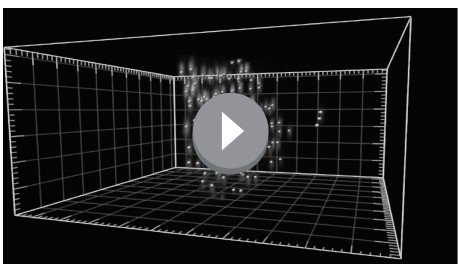

**Video 3.** Length: 16 s; Real Time: 300 s; Frame Rate: 30 fps. eGFP–CENP-A, eGFP–Centrin1. (Deconvolved) Movie of a metaphase cell rendered in 3D overlaid with spot locations (silver spheres) as determined by the kinetochore-tracking assay. Movie rendered used MATLAB and IMARIS. Please also refer to *Figure 2*.

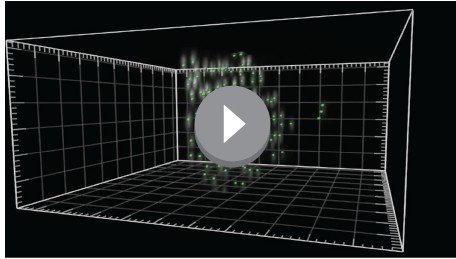

**Video 4.** Length: 16 s; Real Time: 300 s; Frame Rate: 30 fps. eGFP–CENP-A, eGFP–Centrin1. (Deconvolved) Movie of a metaphase cell rendered in 3D overlaid with spot locations (green spheres) and frame-to-frame displacements (green tracks) as determined by the kinetochore-tracking assay. Movie rendered used MATLAB and IMARIS. Please also refer to *Figure 2*.

switch still has a LIDS bias (1.53 compared to a bias of 1.64 for a preceding LIDS), while the (median) time of the coherent run is identical to that following a LIDS (27.1 ± 0.21 s for TIDS, 26.7 ± 0.29 s for LIDS, p=5.5%). However, TIDS take a shorter amount of time to complete (second sister switches) than LIDS, median times 4.05 ± 0.15 s versus 4.31 ± 0.13 s, respectively (p=1.3 x $10^{-4}$). In essence, kinetochore oscillations are robust to sister switching order and the dynamics of the kinetochores is nearly identical after a TIDS and LIDS, except that the inter-sister distance decreases instead of increases, respectively (*Figure 4C,D*). Our observations suggest that the classic choreography (*Figure 1*) represents half of the dynamic with the inter-sister distance relaxing under a LIDS then increasing over the subsequent coherent run – depolymerising K-fibres pulling kinetochores with greater force than polymerising fibres push. However, under a TIDS the inter-sister distance is overstretched and relaxes during the subsequent coherent run – the centromeric spring force increasing the velocity of the trailing sister so that it exceeds that of the lead sister. This TIDS choreography implies that directional switching cannot be triggered by a threshold on the spring tension as overstretching and subsequent relaxation of the spring tension prior to a directional switch are a natural part of metaphase kinetochore oscillations.

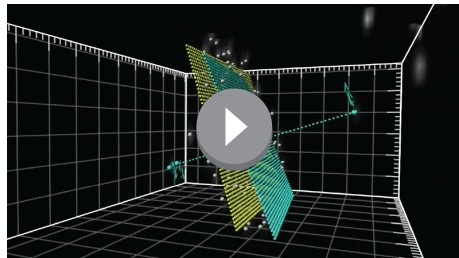

**Video 5.** Length: 7.02 s; Real Time: 300 s; Frame Rate: 30 fps. eGFP–CENP-A, eGFP–Centrin1. (Deconvolved) Movie of a metaphase cell rendered in 3D overlaid with: aligned kinetochore locations (silver spheres); metaphase plate fit (small yellow spheres); spindle poles (blue spheres); spindle pole frame-to-frame displacements (blue tracks); spindle axis and spindle mid-plane (small blue spheres) as determined by the kinetochore-tracking assay. Movie rendered used MATLAB and IMARIS. Please also refer to *Figure 2*.

Kinetochores move in 3D with the kinetochore sister axis being compliant to twist away from the metaphase plate normal (*Figure 2D,E*). The twist and inter-sister distance are in fact inversely correlated (r=–0.13, significant at $p<10^{-200}$), with twist showing inverted profiles over switching events relative to that for the inter-sister distance (*Figure 5*). Specifically, the twist of the sisters falls prior to a LIDS and increases after the switch event, while a TIDS demonstrates the opposite (*Figure 5A*). The average twist increase seen at a LIDS relaxes over the following coherent run (*Figure 5C*), similar to the relaxation in the spring extension seen after a TIDS. This can be explained mechanistically since a high inter-sister stretch, indicative of a high inter-sister tension, aligns the sisters, reducing the twist, while a low stretch, with low tension, allows the twist angle to increase under thermal and mechanical fluctuations. Thus, a negative correlation between twist and inter-sister distance is predicted if there is mechanical compliance in the attachment of the kinetochore to the K-fibres and the centromeric spring.

Examination of the inter-sister distance profiles between consecutive events reveals a key pattern; on average, a TIDS is associated with a lower inter-sister distance than a LIDS 4 s prior to the event regardless of a preceding LIDS or TIDS (*Figure 4B*), a LIDS event having a mean separation above 920 nm (dark blue and grey traces), a TIDS event being below 920 nm (orange and light blue traces). Coupled with the fact that the run is invariant to the nature of the preceding switching event, particularly with regard to timing (the coherent run time to the next directional switch is identical, p=5.5%), this suggests that switching of the lead and trail sisters are governed by clocks. This could be a mechanical timing mechanism associated with the attached K-fibre, and thus force sensitive. In this way, the trailing sister is kept in a polymerising state by the pulling force from the stretched centromeric chromatin (see model in *Figure 6*). This may reflect the ability of the kinetochore to inhibit catastrophe when under tension – as shown by biophysical experiments using purified budding yeast kinetochores (*Akiyoshi et al., 2010*). If that tension falls too low (or fails to build up), a TIDS occurs (as microtubules in the K-fibre undergo catastrophe; *Figure 6*). The same stabilisation mechanism can be invoked to explain switching resolution – during a LIDS the second sister (previously trailing), switches under the loss of the tension in the centromeric chromatin. When tension remains high enough to stabilise the trailing sister, then the lead sister switches because of its clock. The escalating force during a TIDS (after the trailing sister switches) could accelerate the lead sister clock triggering switching of the second sister. Again, this reflects the increase in the rescue rate of in vitro

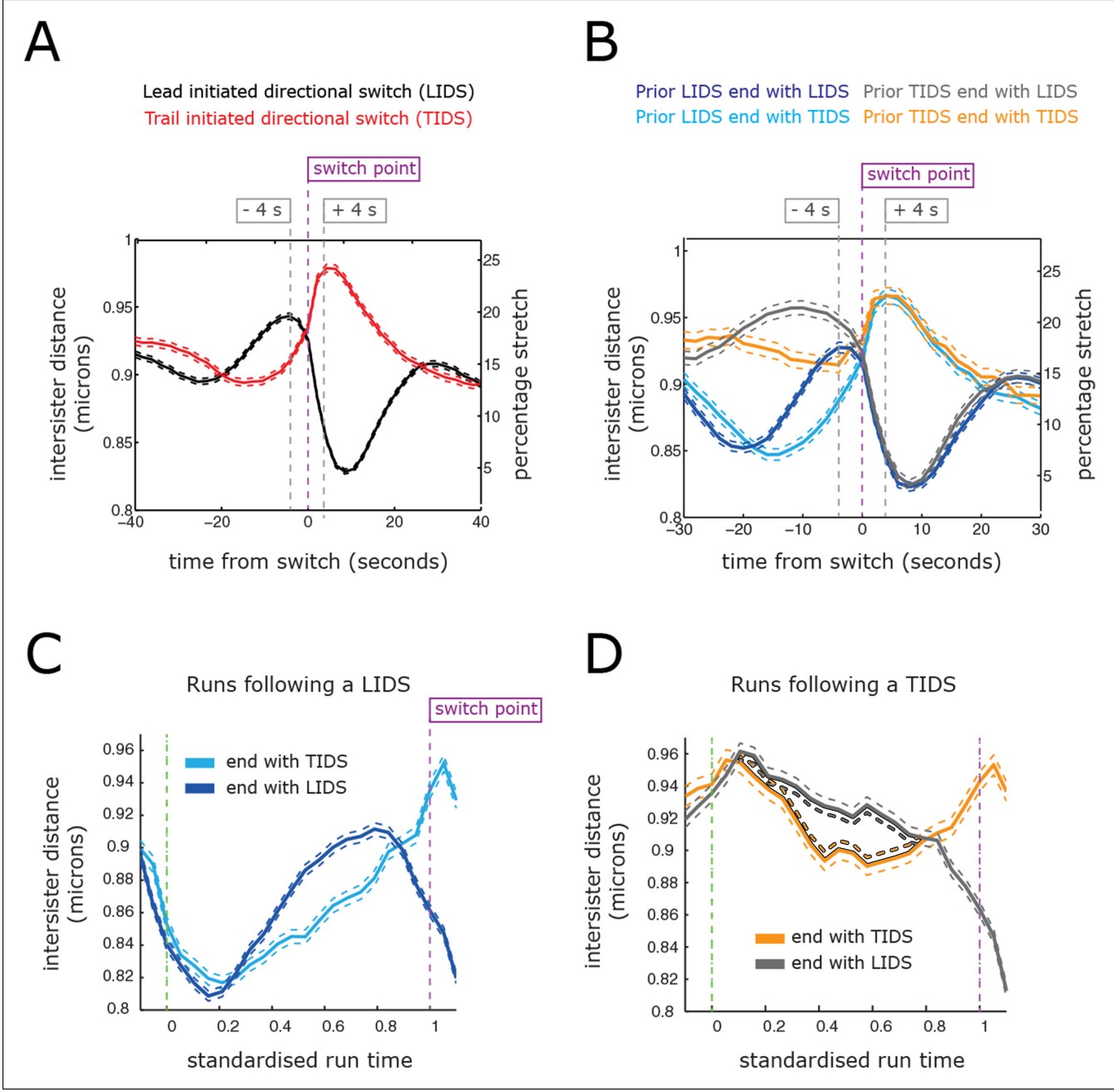

**Figure 4.** Directional switching event signature in the inter-sister distance. (**A**) Time profiles across the first switching event of a directional switch according to type (lead initiated directional switch [LIDS], trail initiated directional switch [TIDS]). Events from trajectories where the lead (LIDS, n=4900; red) or trail (TIDS, n=3143; black) kinetochore switched first were aligned at their modal switching time (time origin, vertical dashed purple line). Solid lines indicate mean values over time, dashed lines ± 1 SEM, smaller than the line thickness where not visible. (**B**) Time profiles across the first sister switch according to type (LIDS, TIDS) separated by prior event (prior LIDS, subsequent LIDS, TIDS n=1536 [dark blue], 936 [pale blue] respectively, prior TIDS and subsequent LIDS, TIDS n=986 [grey], 628 [orange], respectively). In (**A, B**), the percentage stretch relative to the relaxed spring length (determined under nocodazole treatment) is shown on the right axis. Vertical dashed grey lines show 4 s before and 4 s after switch event. (**C**) Inter-sister distance over a standardised average coherent run after a LIDS event categorising runs that exhibit a subsequent LIDS (dark blue) or TIDS (light blue). (**D**) Inter-sister distance after a TIDS categorising runs that exhibit a subsequent LIDS (grey) or TIDS (orange). In (**C, D**), run length is limited to be 6–20 frames inclusive (12–40 s) and rescaled to a standard length of 1 (proportion of run). Sample sizes as (**B**). Dashed lines indicate ± 1 SEM. Vertical green and purple dashed lines indicate the start and end of the coherent run.

The following figure supplements are available for figure 4:

**Figure supplement 1.** Variability in switching times across coherent runs.

*Figure 4 continued on next page*

*Figure 4 continued*

**Figure supplement 2.** Variability in inter-sister distance across switching events.

microtubules attached to kinetochores as shown in *Akiyoshi et al., 2010*, while it emphasises that the two sisters are not symmetrical and the time since their last switch event determines which sister switches.

Although we have detected clear signatures in the switching choreographies (*Figure 4*), these reflect regulatory and mechanical processes of a highly stochastic system. This stochasticity is evident on many scales. First, kinetochores are known to display a range of stochasticity in their trajectories, from near deterministic oscillations to the near random (*Jaqaman et al., 2010*; *Magidson et al., 2011*). Second, the switching times are stochastic; the duration of a coherent run has a large variability with a coefficient of variation (standard deviation [SD]/mean) of 0.45, similar for both runs terminated by an LIDS (mean time and SD 29.6 ± 12.7 s) or a TIDS (29.2 ± 13.6 s, *Figure 4—figure supplement 1*). Thus, although we have invoked a clock mechanism as a switching time regulator, it is inherently stochastic. This stochasticity could stem from both the number and depolymerisation/polymerisation state of individual microtubules that make up the K-fibre. The fraction of microtubules that are in a polymerising state within a K-fibre is highly variable among kinetochores (*Armond et al., 2015),* indicating that growing and shrinking K-fibres are unlikely to be composed of fully coherent microtubules. Third, the signatures in *Figure 4* are a mean behaviour, while variability in the inter-sister stretch throughout the dynamics is in fact large. This can be seen at the population level of trajectories, where the inter-sister distance distributions for LIDS and TIDS only show marginal separation before the switching event (–4 s), are hardly separated at the event, while separation increases after the event (+4 s) (*Figure 4—figure supplement 2A–C*). The inter-sister distance distribution over a switching event is in fact far from bimodal; a mixture of two Gaussian distributions requires the respective means to be separated by at least 2 SDs to be bimodal, the largest we observe is 70% at 4 s post event (*Figure 4—figure supplement 2A–C*). The separation of these distributions does not improve even on further categorising by the prior event (i.e. prior LIDS or TIDS; *Figure 4—figure supplement 2D–F*). Therefore, we have to conclude that the signatures shown in *Figure 4* are not a universal behaviour but only detectable on averaging; that is, the actual switching process is highly stochastic. It may be that analysis of the most deterministic trajectories will reduce this stochasticity in the switching dynamics and signatures. However, directional switching may be a composite process that integrates over multiple signals, that is, tension may not be the only determinant; the stochasticity in our signatures would then be due to measuring only one of these determinants. It remains unknown to what extent the stochasticity in switching time and switching type explains the observed diversity in kinetochore trajectory dynamics, or whether other sources of variability exist.

This paper demonstrates how dynamic and mechanistic insight can be extracted from high-resolution tracking data. Although the leading sister typically initiates directional switching, the reverse switching order is also frequently observed (*Figure 3D*), with directional switching biases, timings and subsequent oscillations remaining robust to such events. By classifying switching order events, we were able to demonstrate clear stereotypical behaviour associated with these events, with both prior (potentially causal) and post-event signatures in the inter-sister distance dynamics. This confirms that event classification is physical and not due to noisy fluctuations in switching times coming from localisation measurement noise.

Our data support a new model of kinetochore oscillations comprising mechanical clocks on both the lead and trailing sister, which likely reflect the time- and force-dependent rescue and catastrophe of the K-fibre microtubules. Our analysis suggests that the degree of stretch of the inter-sister centromeric chromatin is a major determinant in orchestrating this switching; first, a mechanism based on stabilisation of the trailing sister polymerisation state through the centromere tension, effectively slowing the trailing sister clock, and second, a clock on the lead sister that is accelerated under high tension (*Figure 6*). Our data indicate that the standard tension model of sister

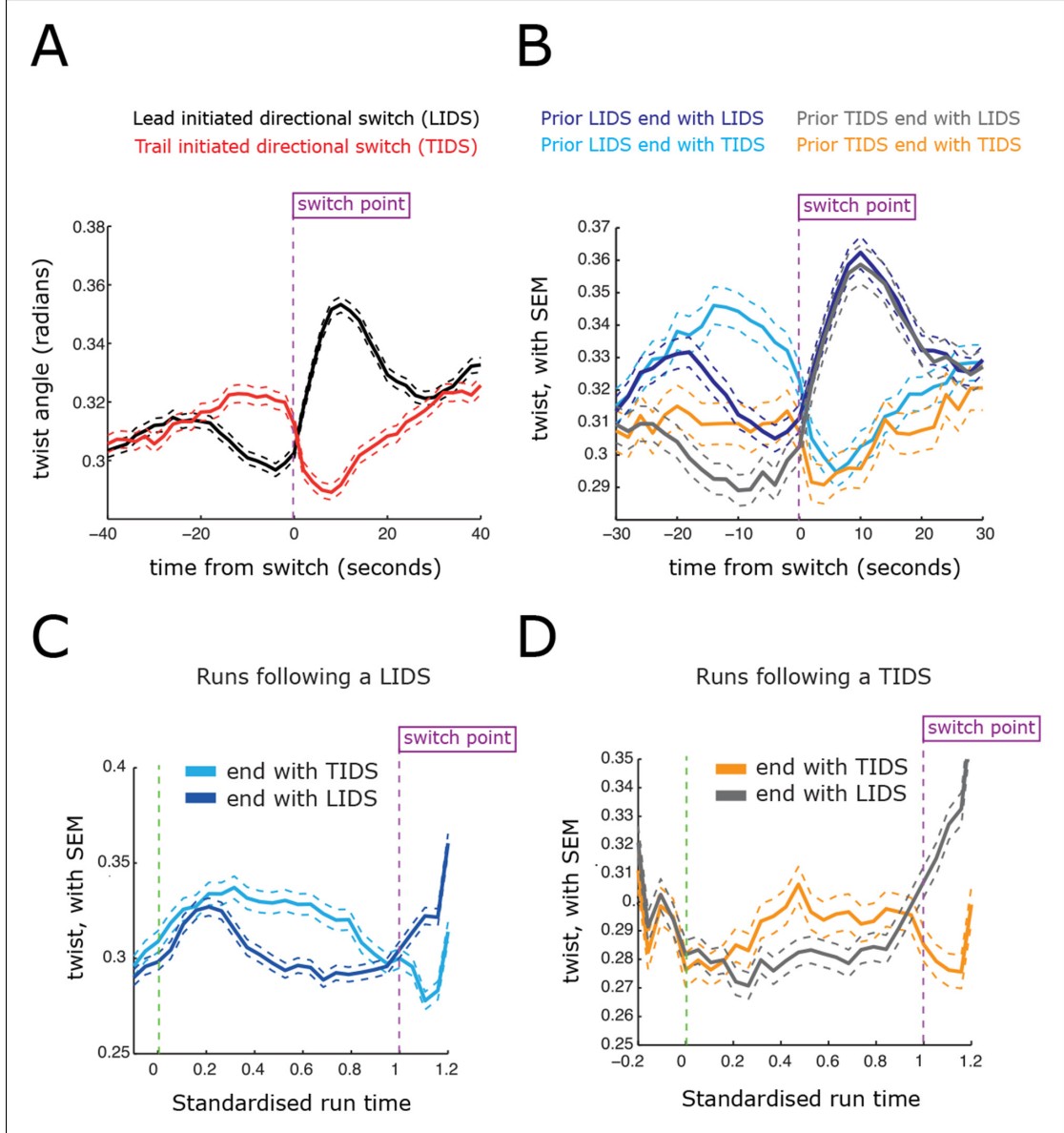

**Figure 5.** Directional switching event signatures in the sister kinetochore twist (angle of the sister axis to the metaphase plate normal). (**A**) Time profiles across the first switching event of a directional switch according to type (lead initiated directional switch [LIDS], trail initiated directional switch [TIDS]). Events from trajectories where the lead (LIDS, n=4900; red) or trail (TIDS, n=3143; black) kinetochore switched first were aligned at their modal switching time (time origin, vertical dotted purple line). Solid lines indicate mean values over time, dashed lines ± 1 SEM, smaller than the line thickness where not visible. (**B**) Time profiles across the first sister switch according to type (LIDS, TIDS) separated by prior event (prior LIDS, subsequent LIDS, TIDS, n=1536 and 936, respectively, prior TIDS and subsequent LIDS, TIDS n=986 and 628, respectively). (**C**) Twist after a LIDS event categorising runs that exhibit a subsequent LIDS (dark blue) or TIDS (light blue). (**D**) Twist after a TIDS categorising runs that exhibit a subsequent LIDS (grey) or TIDS (orange). In (**C, D**), run length is limited to be 6–20 frames inclusive (12–40 s) and rescaled to a standard length of 1 (proportion of run). Sample sizes as (**B**). Dashed lines indicate ± 1 SEM. Vertical green and purple dashed lines indicate the start and end of the run.

kinetochore switching (*Figure 1*) is only able to explain part of the dynamics, while it is incompatible with the overstretch and subsequent relaxation of the inter-sister distance following a TIDS. Thus, kinetochores utilise multiple sensory and fail-safe mechanisms that ensure high-fidelity chromosome organisation within the spindle, despite high levels of stochasticity.

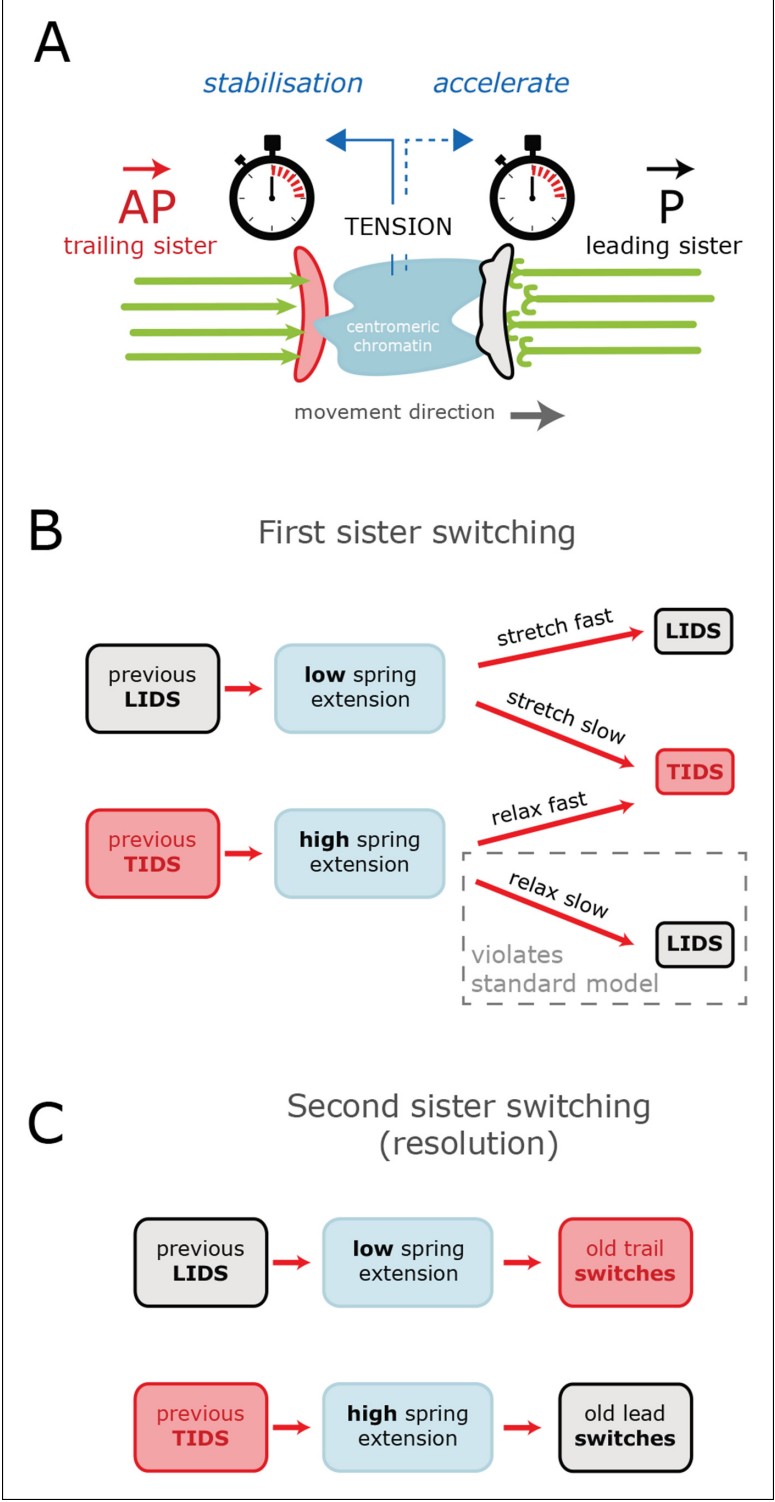

**Figure 6.** Tension-clock model for sister kinetochore directional switching. (**A**) Schematic outline of regulatory mechanisms that control sister kinetochore directional switching. Our data are consistent with the presence of a clock on both the leading (black) and trailing (red) sister kinetochores that sets the time at which a directional switch will occur. We propose that the molecular mechanism for the clock is the rescue/catastrophe frequency of the microtubules (green) within the K-fibre. We note that the kinetics of microtubule dynamic instability in vitro cannot alone explain the timing of events (oscillation period); hence, regulation of the dynamics from the kinetochore/K-fibre structure is key. The clock on the trailing sister kinetochore is force-sensitive, such that tension
*Figure 6 continued on next page*

*Figure 6 continued*

from stretching the centromeric chromatin results in a slow down, thereby reducing the probability that the K-fibre will drive a trail initiated directional switch (TIDS). (B) This mechanism can explain the observed relationship between tension dynamics (in centromeric chromatin) and sequences of lead and/or trail initiated directional switches. Following a LIDS (upper half), the spring extension is low and then stretches over the following run. If tension increases quickly enough, then the trailing K-fibre is stabilised (the clock slows down) and the lead kinetochore switches as the clock runs down. If tension does not build up sufficiently, then the trailing K-fibre will not be stabilised and it will therefore switch because of its clock. Following a TIDS, (lower half), the spring extension is high. The run thus starts with high tension which stabilises the trailing K-fibre. Spring tension relaxes over the run, and if it drops sufficiently, trailing K-fibre stabilisation is lost and the trailing sister switches (TIDS). If tension remains sufficiently high, then the clock on the leading kinetochore initiates the switching event (LIDS). Thus, lead sister initiated switches can occur with falling tension ruling out the standard model (see *Figure 1*). (C) This model can also account for why the second sister switches: following a LIDS, both sisters are in an AP state, reducing the spring tension to a near-zero tension state. This rapidly destabilises the previously trailing sister leading to a switch (AP to P). We suspect that the new lead sister does not switch in this situation because its clock has been re-set (i.e. the K-fibre is new). When a TIDS occurs, we propose that the very high tension generated during the directional switch accelerates the clock on the sister attached to the older K-fibre (previously the lead sister) causing the second sister to switch.

## Materials and methods

### Cell culture and drug treatments

HeLa-K cells stably expressing eGFP-CENP-A (*Jaqaman et al., 2010*) or eGFP-CENP-A/eGFP-Centrin 1 were grown in Dulbecco's modified Eagle's medium (Fisher, UK) containing 10% foetal calf serum (Fisher), 100 µg ml⁻¹ penicillin and 100 µg ml⁻¹ streptomycin maintained in 5% $CO_2$ at 37°C in a humidified incubator. eGFP-CENP-A cells were maintained in 0.1 µg ml⁻¹ puromycin (Fisher). eGFP-CENP-A/eGFP-Centrin 1 cells were maintained in 0.1 µg ml⁻¹ puromycin and 500 µg ml⁻¹ Geneticin (Fisher). To measure the inter-sister distance rest length, cells were treated with 2 µg ml⁻¹ nocodazole (Fisher) for between 16 and 24 hr to depolymerise microtubules. We determined both the mean rest length and the population standard deviation.

### Live cell imaging

Cells were seeded in gridded 35-mm glass bottom dishes (MatTek, Boston, MA) and the media changed to Leibovitz L-15 supplemented with 10% foetal calf serum prior to imaging. Cells were imaged using a 100× 1.4 NA oil objective on a confocal spinning-disk microscope (VOX Ultraview; PerkinElmer, Waltham , MA) with a Hamamatsu ORCA-R2 camera, controlled by Volocity 6.0 (PerkinElmer) running on a Windows 7 64-bit (Microsoft, Redmond, WA) PC (IBM, Armonk, NY). Mitotic cells were first identified using bright-field illumination to minimise phototoxicity. Image stacks (25 z-sections, 0.5 µm apart) were collected every 2 s for 5 min (150 time points per video). Camera pixels were binned 2 × 2, giving an effective pixel size of 138 nm in the lateral direction with a 16-bit per pixel imaging depth. Exposure conditions were set 50 ms per z-slice using a 488-nm laser set to 15% power.

### Image pre-processing

Image pre-processing was performed on an OSX 10.6 Power-Mac (Apple). Time series were exported from their native Volocity format to. OME.TIFFs (The Open Microscopy Environment) using Volocity 6.0 and were then deconvolved with Huygens 4.1 (SVI) using a point spread function (PSF) measured from micro-bead images (using the Huygens 4.1 PSF distiller). Deconvolved images were exported from Huygens to a .r3d format (Applied Precision, Issaquah, WA ) and then read into MAT-LAB (R2012b, MathWorks, Natick, MA) using the loci-tools java library (The Open Microscopy Environment). Images were then stored in a native MATLAB format.

### Kinetochore tracking

Sister kinetochores were detected, aligned, tracked and paired as in the original tracking assay (*Jaqaman et al., 2010*), except that we implemented 3D Gaussian mixture model fitting for

determining sub-pixel spot locations (*Thomann et al., 2002*). In essence, fluorescence from a kineto-chore is modelled as a Gaussian, the fluorescence image then being modelled as a mixture of Gaussians of variable height (intensity). The protocol was tested on both eGFP-CENP-A/eGFP-centrin1 and eGFP-CENP-A cell lines, and gave significantly improved position accuracy compared to the previous centroid-based spot fitting (*Jaqaman et al., 2010*), important here because of the faster time sampling which results in a smaller inter-frame spot displacement. The theoretical accuracy of localisation of a spot's centre ($x,y = \pm 2.8$ nm; $z = \pm 5.7$ nm) was calculated using the total number of photons in the spot, the average background intensity nearby, the full-width half-maximum of the intensity profile in a given coordinate, and the voxel size ($138 \times 138 \times 500$ nm) (*Thompson et al., 2002*). Tracking parameters were identical to *Jaqaman et al. (2010)*, except that the upper limit of the search radius for aligned kinetochores was changed to 0.8. Gap filling as *Jaqaman et al. (2010)* was implemented within the Gaussian MMF tracking. We also filter out cells entering anaphase, removed paired tracks with less than 112 consecutive time points (75% complete) and a small number of tracking errors; this generated a large database of 3D paired trajectory data. Sister kineto-chore movements were calculated relative to a plane fitted through the distribution of sister kinetochore positions. MATLAB software (KiT) is deposited on GitHub and also available on request to ADM.

## Statistical algorithm to extract switching times

We developed a computational algorithm that fits a linear autoregressive statistical model to kineto-chore frame-to-frame displacements that incorporates switching of the driving (constant) term. Specifically displacements are given by

$$\Delta X^1 = c_0 + c(\sigma^1) - aX^1 + bX^2 + N(0, s^2), \Delta X^2 = -c_0 - c(\sigma^2) - aX^2 + bX^1 + N(0, s^2),$$

where $X^1$, $X^2$ are the positions of the sister kinetochores relative to the metaphase plate, the K-MTs lying to the right of $X^1$, left of $X^2$ for sisters 1 and 2, respectively; thus, typically $X^1 > X^2$. The first term ($c_0 + c(\sigma^k)$), k=1,2 are the driving terms with a component that switches between two possible values, positive, $c_+$ corresponding to polymerisation of the K-fibre, or negative, $c_-$ corresponding to depolymerisation. The direction sequence $\sigma^k$ (+,- valued for polymerisation, depolymerisation respectively, with k=1,2 identifying the sister) determines which value is used. Driving terms are of opposite sign between sisters 1 and 2 because the K-fibres lie in opposite directions. If (de)polymeri-sation is turned off, the sisters relax towards the metaphase plate with an inter-sister separation of $2c_0/(b+a)$ which must be positive, thereby constraining the sign of these parameters. The third/fourth terms are relaxation terms allowing kinetochore positions to adjust to the driving term, that is, the relaxation of the inter-sister distance and distance from the metaphase plate. Finally, Gaussian noise is added to model trajectory stochasticity; this will comprise measurement noise, thermal noise and non-thermal ATP-dependent fluctuations (*Weber et al., 2012*). In this model, sisters switch independently (states encoded in $\sigma^k$) from *polymerisation* (+) to *depolymerisation* (-) states (states of their associated K-fibres), and vice versa; the waiting time is exponentially distributed, that is, there is no memory, location or history dependence assumed. The average waiting time to a switch event is dependent on the direction of the other sister; let p be the matrix of switching rates between the 4 sister states ++, +-, -+, -- parametrised by a switching rate out of coherence, p(+-→ ++ or - -)= p(-+→ ++ or - -) and out of incoherence, p(++ → +- or -+)= p(- -→ +- or -+), the sister who switches being chosen at random. There is, therefore, no switching bias intrinsic in the algorithm; biases in the experimental data can thus be detected. This model can produce stochastic saw-tooth oscillations under certain parameter regimes (b,$c_0$>0 and a>b are necessary) (*Figure 3—figure supplement 1*), qualitatively similar to those observed for sister kinetochores. Crucially if the rate of switching out of incoherence is higher than switching out of coherence (coherence of sister movement [same direction] is thus restored quickly), the model produces pseudo-periodic saw-tooth oscillations qualitatively similar to those observed. This model is thus appropriate for detecting switching times as it has the correct type of behaviour.

A Markov chain Monte Carlo (MCMC) algorithm was used to compute the posterior distribution of the parameters and the unknown (hidden) sister states ($\sigma^k$) from each trajectory, that is, sample from the posterior probability density $\pi(a,b,c_0,c_{+/-},s^2,p\ \sigma_t^k|\ X_t^k)$. The MCMC algorithm is based on standard Gibbs and Metropolis–Hastings proposals, and recovered the true values on simulated

data (not shown). We used a prior on the relaxed inter-sister separation of $2c_0/(b+a)$ inferred from a nocodazole experiment (fully depolymerised microtubules) while all other priors are uninformative. Convergence was assessed using multiple runs; a proportion of runs failed to converge despite extending the run time (19%). These trajectories were excluded from the analysis and by visual assessment were typically highly stochastic, suggesting the oscillatory signal was weak and thus lack of convergence was not unlikely. This left a database of 1529 processed paired sister tracks in the eGFP-CENP-A cell line. Each trajectory had sufficient information to fit all the parameters using uninformative priors.

Switch points were determined by identifying coherent runs (classified as a sequence of points where the inferred direction was unchanged for at least five frames). Switch points into and out of a coherent run were matched to determine directional switching events (both sisters switch direction across a directional switch, resulting in a directional switch of the sister pair).

The algorithm was tested for correct determination of switching times on simulated data. Five hundred trajectories were simulated with parameters that gave qualitatively realistic oscillations ($c_+$=30 nm, $c_-$=100 nm, b=0.04, a=0.056, $c_0$=667 nm, $s^2$=1/1000, switching probability per frame 39% [when incoherent], 6.3% [coherent]), a typical trajectory is shown in *Figure 3—figure supplement 1*. The simulation in *Figure 3A* used these same parameters except when incoherent the sister who switched last cannot switch, that is, switches out of coherence always result in a directional switch of the two sisters. Directional switching points were determined with the MCMC algorithm as above and correct LIDS/TIDS calls identified. Accuracy was determined on these 500 trajectory simulations, giving an accuracy of 94%. The inferred lead bias was 1:1, consistent with the original simulation parameters.

## Acknowledgements

The authors thank Jonas Dorn, Chris Smith and Steve Royle for helpful discussions and colleagues of the kinetochore consortium (Gaudenz Danuser, Jason Swedlow, Patrick Meraldi) from which the original kinetochore-tracking assay was developed. They also thank Chris Smith for calculating the accuracy of sub-pixel spot finding.

## Additional information

### Funding

| Funder | Grant reference number | Author |
|--------|------------------------|--------|
| Biotechnology and Biological Sciences Research Council | BB/I021353/1 | Andrew D McAinsh<br>Nigel J Burroughs |
| Wellcome Trust | 106151/Z/14/Z | Andrew D McAinsh |

The funders had no role in study design, data collection and interpretation, or the decision to submit the work for publication.

### Author contributions

NJB, ADM, Conception and design, Analysis and interpretation of data, Drafting or revising the article; EFH, Acquisition of data, Analysis and interpretation of data

### Author ORCIDs

Edward F Harry, http://orcid.org/0000-0003-1914-6432
Andrew D McAinsh, http://orcid.org/0000-0001-6808-0711

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
