## [Decision Letter]

Thank you for submitting your work entitled "Super-resolution kinetochore tracking reveals the mechanisms of human sister kinetochore directional switching" for peer review at *eLife*. Your submission has been favorably evaluated by Vivek Malhotra (Senior editor) and three reviewers, one of whom is a member of our Board of Reviewing Editors.

All three reviewers had the same comment, which was the issue of stochasticity vs. the average behavior. But they disagreed as to how important this was.

Currently, you rely largely on a discussion of the average behavior (Figure 4 and Figure 5). Also Guassian noise is added to the position of kinetochores. In the previous submission this was Brownian motion of the kinetochore. In this version, it was unspecified. In living systems, there are considerably greater "random" fluctuations than just Brownian. A good citation for this can be found in Weber et al., (2012) Nonthermal ATP-dependent fluctuations contribute to the in vivo motion of chromosomal loci. The non-thermal fluctuations come from the motors, chaperones, ATP machines etc. that are constantly jiggling everything in the cell. Thus the kinetochore is most likely being jostled with a magnitude much greater than Brownian, but with random trajectory. Just examining the trajectory is insufficient to distinguish whether the motion is truly random (Brownian) or these non-thermal energy-dependent events that appear to be randomly orientated.

After discussion between the three reviewers, we have decided that we would like to publish the manuscript, but would like you to be much more explicit and extensively address the issue of average behaviour versus stochasticity. In particular, a discussion of how much individual traces might depart from this average behavior is absent and could be added. At the very least the spread of data should be indicated (see point 2 below). I think that a discussion of this issue would really add to the results here, right now the model put forward at the end in Figure 6 is one of a clock that runs and determines the switching dynamics, leaving the reader with the impression that this is a rather deterministic process. In reality switching is governed by stochastic events, the 'clock' probably is the characteristic timescale of some biochemical reaction and the variance will be as large as the characteristic timescale. Thus, the variance in the behavior of individual switching events will be large, (probably) consistent with the data here. In other words you need to clearly highlight the stochastic nature of the entire process (after all, the switching order is highly stochastic).

Other comments:

1) By examining the shape of TIDS or LIDS curves, respectively in Figure 4, it is clear that the original TIDS traces (before averaging) must have had a very large spread. It would be interesting if you commented on the standard deviation of the curves. In particular, it would be interesting if the standard deviation after a switching event is very small after a TIDS or LIDS switching event. This would further support your claim that the behavior after a switch is very similar. Moreover, an examination of the spread in the data may indicate variables that could be of importance. For instance, is the spread Gaussian or does it show a bi-modal distribution. If the behavior after a TIDS or a LIDS is different, you may expect the distribution of curves to be bi-modal with a peak centered at each type of behavior.

2) The shape of the curve is remarkably similar between a TIDS and LIDS event (Figure 4), just slightly smaller and inverted. Also, the twist shows a similar difference (Figure 5). Is type of behavior anticipated by the model proposed by you?

3) In your proposed model the speed of relaxation or stretch depends on whether the previous event was a TIDS or LIDS (alternatively, if there is high or low spring tension). Did you analyze whether relaxation times of individual runs correlated with either previous event type or spring tension?

4) You propose that the clock timing in your model is set by the rescue/catastrophe frequency of microtubules. Are there previous studies or do you have evidence to expect that such an underlying mechanism would lead to a switching frequency of 36 seconds?

[Editors' note: further revisions were requested prior to acceptance, as described below.]

Thank you for resubmitting your work entitled "Super-resolution kinetochore tracking reveals the mechanisms of human sister kinetochore directional switching" for further consideration at *eLife*. Your revised article has been favorably evaluated by Vivek Malhotra (Senior editor) and a Reviewing editor. The manuscript has been improved but there are some remaining issues that need to be addressed before acceptance, as outlined below:

As you know from the reviewers’ comments, the issue about stochastic behaviour versus average behaviour was a crucial issue. We still feel that needs better clarification. We think this important, because given the unanimous feeling of all reviewers, I don't think your paper will get the attention it deserves unless it is cleared up.

Figure 4—figure supplement 1 needs additional clarification. What is not immediately clear is what times A, B, and C refer to. In the legend, where you appear to refer to Figure 3 (and not Figure 4) you say that A is at start of event, B is 4s prior and C is at event. How does this relate to the times shown in Figure 4 or Figure 4? What would be needed here is a supplemental figure that indicates the distributions for Figure 4. Please generate one that shows the distributions for the intersister distances at three times, at the switch point and +/- ~5 seconds, to get a feeling how much these distributions of which the means are shown in 4A and B overlap.

In addition, the corresponding paragraph of discussion (Results and Discussion, ninth paragraph) is a bit weak and all over the place. Please go through this and try and be more precise in your arguments.

---

## [Author Response]

*[…] After discussion between the three reviewers, we have decided that we would like to publish the manuscript, but would like you to be much more explicit and extensively address the issue of average behaviour versus stochasticity. In particular, a discussion of how much individual traces might depart from this average behavior is absent and could be added. At the very least the spread of data should be indicated (see point 2 below). I think that a discussion of this issue would really add to the results here, right now the model put forward at the end in Figure 6 is one of a clock that runs and determines the switching dynamics, leaving the reader with the impression that this is a rather deterministic process. In reality switching is governed by stochastic events, the 'clock' probably is the characteristic timescale of some biochemical reaction and the variance will be as large as the characteristic timescale. Thus, the variance in the behavior of individual switching events will be large, (probably) consistent with the data here. In other words you need to clearly highlight the stochastic nature of the entire process (after all, the switching order is highly stochastic).*

We are really pleased that the reviewers found our paper to be important and very much appreciate the constructive comments on our work. We have altered the manuscript as suggested. Specifically we have:

1) Added a comment about thermal and nonthermal components to the noise, in the first paragraph of the subsection “Statistical algorithm to extract switching times" Statistical algorithm to extract switching times” and have added the suggested citation (Weber et al., PNAS, 2012);

2) Included a discussion of the stochasticity in the system as opposed to the average behavior, in the ninth paragraph of the Results and Discussion. Further, we have added Figure 4—figure supplement 1 showing the distributions of the inter-sister distance at 3 times throughout a coherent run. This shows that the distributions have a large variance and are approximately Gaussian. Unfortunately, there is no bimodality apparent in these distributions. We discuss this in the aforementioned paragraph. We have also included a plot of distribution of switching times to highlight the stochasticity in this variable (see Figure 4—figure supplement 1).

*Other comments:*

*1) By examining the shape of TIDS or LIDS curves, respectively in Figure 4, it is clear that the original TIDS traces (before averaging) must have had a very large spread. It would be interesting if you commented on the standard deviation of the curves. In particular, it would be interesting if the standard deviation after a switching event is very small after a TIDS or LIDS switching event. This would further support your claim that the behavior after a switch is very similar. Moreover, an examination of the spread in the data may indicate variables that could be of importance. For instance, is the spread Gaussian or does it show a bi-modal distribution. If the behavior after a TIDS or a LIDS is different, you may expect the distribution of curves to be bi-modal with a peak centered at each type of behavior.*

1) See point 2 above. The distribution standard deviations are large and only alter by up to 10% throughout switching so we do not feel that commenting on these changes adds much to the understanding.

*2) The shape of the curve is remarkably similar between a TIDS and LIDS event (Figure 4), just slightly smaller and inverted. Also, the twist shows a similar difference (Figure 5). Is type of behavior anticipated by the model proposed by you?*

We have now emphasized how the negative correlation between twist and intersister distance reflects mechanical processes of the kinetochore, in the seventh paragraph of the Results and Discussion. The inverse patterns of LIDS and TIDS are observational and we do not believe have a mechanistic underpinning.

*3) In your proposed model the speed of relaxation or stretch depends on whether the previous event was a TIDS or LIDS (alternatively, if there is high or low spring tension). Did you analyze whether relaxation times of individual runs correlated with either previous event type or spring tension?*

The reviewers are touching on the memory aspects of switching. Our preliminary analysis suggests that memory exists. A manuscript is in preparation.

*4) You propose that the clock timing in your model is set by the rescue/catastrophe frequency of microtubules. Are there previous studies or do you have evidence to expect that such an underlying mechanism would lead to a switching frequency of 36 seconds?*

There is no previous data that could allow one to predict that K-fibre rescue/catastrophe dynamics would give rise to a ½ period of 36 s. We have added a sentence to make this point in the legend to Figure 6. It is likely that this arises from an interplay of all the microtubule binding, and kinetochore proteins. We have recently developed assays that allow microtubule polymerization dynamics to be probed at single kinetochores (Armond et al., J. Cell Sci., 2015), We hope that future work analyzing such dynamics throughout switching events – and how this is dependent on kinetochore components – will allow us to begin answering the important question raised by the reviewers.

[Editors’ note: the author responses to the previous round of peer review follow.]

*As you know from the reviewers’ comments, the issue about stochastic behaviour versus average behaviour was a crucial issue. We still feel that needs better clarification. We think this important, because given the unanimous feeling of all reviewers, I don't think your paper will get the attention it deserves unless it is cleared up.*

We are very sorry that our initial effort to address the concerns raised by the reviewers was not sufficiently clear. We do agree that this is an important issue and have now rewritten the text, added additional measures to quantify the stochasticity and added additional figures to clarify the issue of stochastic behaviour as follows below.

*Figure 4—figure supplement 1 needs additional clarification. What is not immediately clear is what times A, B, and C refer to. In the legend, where you appear to refer to Figure 3 (and not Figure 4) you say that A is at start of event, B is 4s prior and C is at event. How does this relate to the times shown in Figure 4 or Figure 4? What would be needed here is a supplemental figure that indicates the distributions for Figure 4. Please generate one that shows the distributions for the intersister distances at three times, at the switch point and +/- ~5 seconds, to get a feeling how much these distributions of which the means are shown in 4A and B overlap.*

As suggested, we have now included histograms of the inter-sister distance at the switch point, -4 sec and + 4 sec from the switch point (see new Figure 4—figure supplement 2). We need to use the 4 sec, rather than 5 sec, time point due to the 2 sec sampling rate used in the image acquisition. We have included histograms that map directly to both Figure 4. To aid the reader we have also indicated these time points in Figure 4/B (vertical grey dotted lines) and improved the figure annotation compared to the previous version.

*In addition, the corresponding paragraph of discussion (Results and Discussion, ninth paragraph) is a bit weak and all over the place. Please go through this and try and be more precise in your arguments.*

We have completely re-structured this paragraph to improve the clarity of our arguments and to clearly outline the different levels of stochasticity in the dynamics. We have also outlined the possible origin of this stochasticity. We have now added a sentence to explicitly state that the inter-sister distance distribution is not bimodal and further quantify how far it is from bimodality (original comment 1 from reviewers):

"The inter-sister distance distribution over a switching event is in fact far from bimodal; a mixture of two Gaussian distributions requires the respective means to be separated by at least two standard deviations to be bimodal, the largest we observe is 70% at 4 s post event, Figure 4—figure supplement 2.”